# The Bioactive Phenolic Agents Diaryl Ether CVB2-61 and Diarylheptanoid CVB4-57 as Connexin Hemichannel Blockers

**DOI:** 10.3390/ph15101173

**Published:** 2022-09-21

**Authors:** Anne Dierks, Corinne Vanucci-Bacqué, Anne-Marie Schäfer, Tina Lehrich, Frederike Ruhe, Patrik Schadzek, Florence Bedos-Belval, Anaclet Ngezahayo

**Affiliations:** 1Department of Cell Physiology and Biophysics, Institute of Cell Biology and Biophysics, Leibniz University Hannover, 30419 Hannover, Germany; 2Laboratoire de Synthèse et Physicochimie des Molécules d’Intérêt Biologique, Université Paul Sabatier, CEDEX 9, 31062 Toulouse, France; 3CNRS, UMR 5068, Laboratoire de Synthèse et Physicochimie des Molécules d’Intérêt Biologique, CEDEX 9, 31062 Toulouse, France; 4Center for Systems Neuroscience (ZSN), University of Veterinary Medicine Hannover, 30559 Hannover, Germany

**Keywords:** connexin channels, inflammation signals, dye uptake, transepithelial electrical resistance (TEER), polyphenols, curcuminoids, Calu-3 cells

## Abstract

Inflammation mediators enhance the activity of connexin (Cx) hemichannels, especially in the epithelial and endothelial tissues. As potential release routes for injury signals, such as (oligo)nucleotides, Cx hemichannels may contribute to long-lasting inflammation. Specific inhibition of Cx hemichannels may therefore be a mode of prevention and treatment of long-lasting, chronic sterile inflammation. The activity of Cx hemichannels was analysed in N2A and HeLa cells transfected with human Cx26 and Cx46 as well as in Calu-3 cells, using dye uptake as functional assay. Moreover, the possible impacts of the bioactive phenolic agents CVB2-61 and CVB4-57 on the barrier function of epithelial cells was analysed using Calu-3 cells. Both agents inhibited the dye uptake in N2A cells expressing Cx26 (>5 µM) and Cx46 (>20 µM). In Calu-3 cells, CVB2-61 and CVB4-57 reversibly inhibited the dye uptake at concentrations as low as 5 µM, without affecting the gap junction communication and barrier function, even at concentrations of 20 µM. While CVB2-61 or CVB4-57 maintained a reduced dye uptake in Calu-3 cells, an enhancement of the dye uptake in response to the stimulation of adenosine signalling was still observed after removal of the agents. The report shows that CVB2-61 and CVB4-57 reversibly block Cx hemichannels. Deciphering the mechanisms of the interactions of these agents with Cx hemichannels could allow further development of phenolic compounds to target Cx hemichannels for better and safer treatment of pathologies that involve Cx hemichannels.

## 1. Introduction

Connexins (Cx) are membrane proteins whose most recognised function is the formation of gap junction channels that directly connect the cytoplasmic spaces of adjacent cells in tissue [1,2]. The Cx gap junction channels are large enough to allow the exchange of ions and metabolites, such as (oligo)nucleotides and peptides, up to 1.5 kDa [1,3]. Cxs also form unopposed hemichannels in the membranes of individual cells [2]. In most tissues, the activity of the Cx hemichannels is maintained at a very low level by the docking between adjacent cells that reduces the density of unopposed Cx hemichannels. Moreover, the interstitial Ca^2+^ concentration of about 2 mM strongly reduces the open probability of Cx hemichannels [4]. Due to their large pores, Cx hemichannels allow the influx of solutes from the extracellular milieu as well as the release of intracellular molecules, which may act as injury signals in the tissue [2].

Increased activity of Cx hemichannels has been involved in inflammation, the hallmark of various pathologies, such as atherosclerosis, neurodegenerative diseases, respiratory stress, and genetic-related disorders in different organs, as well as the age-related disabilities commonly referred to as inflamm-aging [5,6,7,8,9]. Cx hemichannels have also been reported as being involved in tumour development [10]. The mechanisms that lead to the increased activity of Cx hemichannels under the mentioned pathological conditions are still a matter of ongoing research. Due to experimental convenience, an increased expression of mainly Cx43 has been extensively reported [11,12,13]. Likewise, the link between the activity of Cx hemichannels and the pathological status is still poorly understood. For tumours, a role of Cx hemichannels as release routes for anti-oxidant [14] and survival mediators, such as bisphosphates [15], has been proposed. In the case of inflammation, the enhancement of Cx hemichannels and a resulting increase in the release of injury signals, such as (oligo)nucleotides, which may continue to trigger inflammation even after infection decay or injury repair, is one of the hypotheses that are often proposed [16,17]. In addition to the clarification of the mechanisms leading to the increased activity of Cx hemichannels, there is considerable research targeting the development of agents that block Cx hemichannels to prevent or treat various diseases [18]. An ideal blocker would be expected to: (i) selectively target the Cx hemichannels and not the gap junction channels, (ii) be isoform-specific, and (iii) not alter the tissue physiology.

Inflammation is a response to invader microbes and the substances released from them commonly known as pathogen-associated molecular patterns (PAMPs). Inflammation is also triggered when tissue is injured. In this latter case, injured cells release into tissue intracellular molecules, such as nucleotides, known as tissue injury signals, commonly called damage-associated molecular patterns (DAMPs) [19]. After binding to so-called toll-like receptors (TLR), mainly in the cell membranes of epithelial cells, the invading microorganisms and their released PAMPs may induce injury-like responses in the epithelial cells, resulting in the release of DAMPs and the activation of cytokine synthesis and secretion [20]. The released DAMPs and cytokines can act as chemoattractants and chemostimulants for immune cells, which trigger inflammation as a starting point for the removal of the invader or for injury healing [20,21]. 

Enhancement of the activity of Cx hemichannels in epithelial cells in response to PAMPs and DAMPs, as recently shown [12,22], may correlate with an increased release of injury signals during inflammatory events. An increased and possibly persistent enhancement of Cx hemichannels may therefore represent a detrimental response of epithelial cells to invaders and PAMPs that would continue to release DAMPs even after the decay of infection.

Polyphenols and curcuminoids are known for their anti-inflammatory activity [23]. These natural products, found in food and beverages as well as in nutrient supplements, present low toxicity and are currently tested in clinical usage mostly as complements for other anti-inflammation pharmaceutics [24,25]. In animal models, polyphenols and curcuminoids inhibit the reactive oxygen species production or the expression and release of inflammatory cytokines by innate immune system cells [26,27,28]. Although these natural products exhibit poor pharmacokinetic properties, they represent a good starting point for the design of biological active agents. In the present article, we report the study of CVB2-61, a phenolic diarylether, which is a dimer of a vanillin derivate with antiatherogenic properties [6] and CVB4-57, a diarylheptanoid and a curcuminoid analogue, as specific agents to target Cx hemichannels.

## 2. Results

### 2.1. Synthesis of the Bioactive Phenolic Agents CVB2-61 and CVB4-57

CVB2-61 (Figure 1a) was synthesised as previously described [29]. CVB4-57 was obtained in three synthesis steps, starting from piperazine, involving two successive N-alkylations with (4-(2-bromoethyl)phenoxy)(tert-butyl)dimethylsilane and 2-bromo-4-(bromomethyl)-1-methoxybenzene, followed by tetrabutylammonium fluoride (TBAF) deprotection of the phenol group, as depicted in Figure 1b.

### 2.2. Connexin Expression in N2A and HeLa Cells

Molecular cloning was used to generate IRES vectors containing the DNA sequence of the Cx isoform Cx26 or Cx46 and GFP. We used these two Cx isoforms for their known function/ability to form Cx hemichannels in vivo in tissues such as the ocular lens, the skin, and the inner ear [30,31]. Additionally, a crystal structure model of Cx26 [32] is available and gives valuable information for further analysis and simulation processes [33] in common studies of the interaction mechanisms of the compounds with Cx hemichannels. To analyse their capacity to form Cx channels, the constructs were expressed in N2A and HeLa cells (Appendix A) that did not express endogenous Cxs. The IRES vector allowed a concomitant expression of GFP and Cxs. However, these two proteins were not linked together. The presence of GFP in the cytoplasmic space was used as control for a successful transfection and expression of the Cxs but did not interfere with the Cx channels’ functionality. Confocal laser scanning microscopy showed a reliable expression of the proteins in the cells, with a transfection efficiency of about 35%. A similar transfection efficiency was also observed in cells transfected with a vector containing only the GFP gene (Appendix A). These latter were used as control in further analysis of the functionality of the Cx hemichannels. The expression of the Cxs was further verified by immunofluorescence staining (Appendix A). As shown in Appendix A, the Cxs were expressed and exported to the cell membrane. In some cells, Cx associations were found at the cell-to-cell contact and the cell border (Appendix A; red arrows), where they presumably formed cell-to-cell gap junction channels or Cx hemichannels. In other cells, a cytoplasmic localisation, probably in the endoplasmic reticulum (ER) and Golgi apparatus of the Cxs, was observed (Appendix A; white arrows).

### 2.3. Functionality of Expressed Cx Hemichannels

In order to analyse the functionality of the Cx hemichannels formed in the membranes of transfected cells, we performed dye-uptake experiments [11,34,35,36]. The functionality of the Cx hemichannels was recognised as an accelerated increase in fluorescence intensity of the dye ethidium bromide (Etd) in the cells in response to the removal of external Ca^2+^, as compared to control conditions with 2 mM external Ca^2+^ (Figure 2a). The rate of dye uptake was quantified to compare cells expressing the Cx variants with cells expressing GFP alone. The fluorescence intensity of Etd was measured in the cells and plotted vs the amount of time for which the cells were incubated with or without external Ca^2+^. A linear regression was applied on the measured curve. The rate of dye uptake in the presence or absence of external Ca^2+^ was estimated as the slope of the linear regression in Ca^2+^-containing and Ca^2+^-free sectors of the experiment (Figure 2a). The rate of dye uptake in cells expressing the Cx variants and those expressing GFP alone was low when external Ca^2+^ (2 mM) was present. The removal of external Ca^2+^ showed an increased rate of dye uptake in cells expressing the Cx variants compared to cells expressing GFP alone (Figure 2b and Appendix Ab). It is worth noting that the dye-uptake rate in cells expressing GFP without Cx variants was similar to that measured in non-transfected cells (Appendix Aa). As shown in Figure 2b, the rate of Etd uptake in cells expressing Cx variants together with GFP was significantly higher compared to that estimated in cells expressing GFP alone, with 3.1 ± 0.4-fold hCx26 (*p* < 0.001) and 4.1 ± 0.5-fold hCx46 (*p* < 0.001) in N2A cells (Figure 2b; for HeLa cells: Appendix Ab). This observation constitutes evidence that even if the open probability of the hemichannels were reduced by external Ca^2+^, the presence of Cx hemichannels in the membrane allowed an uptake of Etd in the N2A cells that was significantly higher in cells expressing Cxs compared to cells expressing GFP alone (Figure 2b). The N2A cells expressing the respective Cx provide a good basis for testing the capacity of different compounds to inhibit Cx hemichannel activity.

### 2.4. Inhibition of Cx Hemichannels

CVB2-61 and CVB4-57 were tested on cells expressing hCx26 and hCx46 (Figure 3 and Appendix A). In presence of 5 µM CBV2-61, the dye uptake in cells expressing hCx26 decreased to 60 ± 6%. An increase in the CBV2-61 concentration to 20 µM did not result in a significant increase in the effect. However, the statistical significance of the reduction in the dye-uptake rate (*p* < 0.05 for 5 µM; *p* < 0.01 for 10 µM; *p* < 0.001 for 20 µM) was strengthened in comparison to the control experiment without CVB2-61 (Figure 3a). For CVB4-57, a significant reduction in the dye-uptake rate (79 ± 17%) was observed even at 1 µM (*p* < 0.05) in cells expressing hCx26 (Figure 3b). Increasing the concentration to 5 µM reinforced the reduction to 50 ± 12% (*p* < 0.01). A further increase in the CVB4-57 to 20 µM induced a further reduction in the dye-uptake rate (34 ± 2%, *p* < 0.001) (Figure 3b). In N2A cells expressing Cx46, a reduction in the dye uptake of about 72 ± 3% (*p* < 0.001) was only observed at the high concentration of 20 µM for both CVB2-61 and CVB4-57 (Figure 3c,d).

In addition to N2A cells, we studied the effects of CVB2-61 and CVB4-57 on Cx channels in Calu-3 cells. In contrast to N2A cells, Calu-3 cells endogenously expressed Cxs. Calu-3 cells are epithelial cells of the human respiratory airways forming a good barrier in vitro [37], and are therefore considered an adequate model for investigating the physiological functions of the lung epithelium [11,38], as well as for the development of therapeutics for various lung diseases [39]. The ability of CVB2-61 and CBV4-57 to inhibit Cx hemichannels in Calu-3 cells was studied (Figure 4 and Appendix A) in reference to previously published results, which showed that inflammatory signals, such as adenosine, enhanced the activity of Cx26 hemichannels in Calu-3 cells [11]. CVB2-61 or CVB4-57 (5 µM) was added to the cells at the same time as the removal of external Ca^2+^. An inhibition of the dye uptake was then noticed (Figure 4 and Appendix A). This effect was observed in cells that were cultivated for 24 h in presence of the adenosine analogue 5-N-Ethylcarboxamidoadenosine (NECA) as well as in cells cultivated under control conditions (Figure 4 and Appendix A). The repression of the dye uptake was due to the presence of CVB2-61 and CVB4-57, since the removal of the agents during the dye-uptake experiments correlated with an immediate acceleration of the dye-uptake rate. In addition, the enhanced rate of dye uptake related to cultivation of the cells under stimulation of adenosine receptors using NECA was still observable after removal of the compounds CVB2-61 and CVB4-57 (Figure 4 and Appendix A).

The above-presented results show that when applied for the short period of the dye-uptake experiment, the response of the Calu-3 cells did not differ between the compounds CVB2-61 and CVB4-57. When the compounds were applied for a long application period (24 h), the response induced by CVB2-61 differed slightly from that induced by CVB4-57. In cells cultivated in the presence of CVB2-61, the increase in the dye-uptake rate observed after the suppression of both external Ca^2+^ and CVB2-61 was reinforced compared to cells cultivated under control conditions (Figure 4a and Appendix A). An additional enhancement was further observed in cells cultivated in the presence of CVB2-61 and NECA (Figure 4a and Appendix A). In contrast, for cells cultivated in the presence of CVB4-57, the increase was similar to that found in cells cultivated under control conditions (Figure 4b). Correspondingly, the enhancement of the dye-uptake rate due to NECA was not affected by the presence of the CVB4-57 during the cultivation compared to cells cultivated without CVB4-57 and in the presence of NECA (Figure 4b). These different effects of CVB2-61 and CVB4-57 should be exploited to further study how they block Cx hemichannels and their overall effects in the cell.

Many Cx channel inhibitors, such as glycyrrhetinic acid (GA) and its derivate carbenoxolone (CBX), affect both the Cx hemichannels and the gap junction channels [40]. In addition, it was shown that GA affected the barrier function of epithelial cells [41]. Our data also show that CBX altered the barrier function at concentrations commonly used to block Cx channels (Figure 5a). In contrast, we found that neither CVB2-61 nor CVB4-57 altered the barrier function of Calu-3 monolayers cultivated in transwell inserts, as shown by measuring the transepithelial electrical resistance (TEER) by impedance spectroscopy (Figure 5a). Moreover, the agents inhibited the dye uptake, meaning they blocked the Cx hemichannels without altering the gap junction related dye transfer. Using the gold-nanoparticle-mediated laser perforation/dye transfer (GNOME-LP/DT) method as described previously [11,42], we found that neither agent affected the degree of gap junction coupling, either when applied spontaneously or when applied during cultivation for 24 h (Figure 5b). Taken together, the above reported results suggest that CVB2-61 and CVB4-57 could block Cx hemichannels with minimal impact on the epithelial physiology.

In summary, the present study shows that the agents CVB2-61 and CVB4-57 are able to reversibly block Cx hemichannels in a Cx expression system as well as in native cells without affecting the activity of the gap junction channels. In comparison, Cx26 hemichannels are more sensitive to both agents compared to Cx46.

## 3. Discussion

Evidence has accumulated showing that inflammation inducers such as the lipopolysaccharide (LPS) of gram negative bacteria, and injury signals such as adenosine enhance the activity of Cx hemichannels in different tissues, especially epithelial layer [11,12,13]. Since hemichannels are permeable to molecules that act as injury signals, the enhancement of the activity of Cx hemichannels may represent a detrimental response of cells to inflammation inducers. They may start and sustain a long-lasting inflammation that persists after the infection decays, with the risk of becoming a chronic sterile inflammation [6,13], the hallmark of various but unrelated pathological situations, such as chronic obstructive pulmonary disease (COPD), asthma in the respiratory system [43], or neurological degenerative diseases [6]. It was therefore proposed that compounds able to block the hemichannels could be used for the prevention and treatment of these different pathologies [18]. As Cx hemichannels are also involved in cancer development [10], inhibitors of Cx hemichannels could be of interest for cancer therapy [44]. Finally, for a low general toxicity, the ideal compound should target the Cx hemichannels specifically without affecting the gap junction channels or altering the tissue physiology.

So far, various small molecules have been tested as blockers of Cx hemichannels, including the antimalarial agents quinine and mefloquine; non-steroidal anti-inflammatory drugs such as fenamates; the anaesthetic drugs propofol and ketamine; glycyrrhetinic acid (GA) and its derivatives, such as carbenoxolone (CBX); alcohols such as heptanol and octanol; 2-aminoethoxydiphenyl borate; and mimetic peptides of Cx extracellular loop sequences, anti-Cx antibodies, and aminoglycoside antibiotics and their derivatives without antibiotic effects. Many small molecules and mimetic peptides mentioned above, as well as anti-Cx antibodies, only poorly distinguished between the gap junction channels and the Cx hemichannels [45,46]. However, the small molecule boldine [47] and the mimetic peptide Gap19 [48], as well as the aminoglycoside antibiotics [49,50], were shown to block Cx43 hemichannels. The mimetic peptide Gap19, designed on the basis of the Cx43 intracellular loop sequence, needs to enter the cells to function. Boldine is an alkaloid tested on Cx43 that may function without entering into the cells. Among the antibodies, a potent anti-Cx26 antibody capable of blocking Cx26 hemichannels without affecting the gap junction channels was generated [51]. With respect to antibiotics, the compound 12i, a derivative of kanamycin A, showed a selectivity toward Cx43 hemichannels compared to Cx26 hemichannels [49,52]. The previously mentioned anti-Cx26 antibody, the Gap19 mimetic peptide, and the small molecule boldine, as well as the antibiotics derivatives, show the ongoing development of compounds capable of specifically inhibiting Cx hemichannels. In the present report, we studied two agents, CVB2-61 and CVB4-57, which were able to reversibly block human Cx hemichannels in an expression system as well as in native cells (Figure 3 and Figure 4). In Calu-3 cells (human lung epithelial cells), which are naturally expressing Cx [11], the two agents were found to not affect the gap junction coupling (Figure 5b), suggesting that they may specifically target Cx hemichannels. Moreover, Cx26 hemichannels were inhibited by the agents even at concentrations as low as 5 µM, while a higher concentration (20 µM) was needed to inhibit Cx46 hemichannels (Figure 3). Finally, the two agents did not appear to alter the barrier function of the epithelial cells, an important physiological aspect of the epithelium (Figure 5a).

CVB2-61 [29] is a dimer constituted by two benzothiazole-vanillin units linked together by an ether oxide bridge (Figure 1a). Due to its antioxidant, antiradical, and antiangiogenic properties, CVB2-61 is a good drug-candidate for preventing atherosclerosis [29]. CVB4-57 is an original analogue of the curcuminoids, which are known for their anti-inflammatory properties [53,54]. CVB4-57 exhibits a phenolic moiety, which may confer its radical quenching properties [55]. A piperazine core was inserted between the two aromatic moieties to replace the heptanoid chain in CVB4-57, ensuring the agent’s rigidity, and finally, two nitrogen atoms were added for potential H-bond interactions with Cx amino acids (Figure 1b). Since piperazine has a well-known pharmacophore [37], a low cytotoxicity for CVB4-57 was expected.

The data presented here show that both CVB2-61 and CVB4-57, at the concentrations used, may selectively affect the Cx hemichannels without disturbing the Cx gap junction channels, either in the opened or in the closed formation (Figure 5b). Moreover, Cx26 was found to be more sensitive to the agents than Cx46 (Figure 3), suggesting that a Cx isoform specificity may be achieved. Furthermore, in Calu-3 cells treated simultaneously with the respective agent and the adenosine receptor agonist NECA for 24 hours, both agents maintained a reduced dye-uptake rate. After their removal from the cells during the experiments, the activity of the hemichannels was released and the NECA-induced enhancement of the rate of dye uptake [11] was still observed, compared to cells cultivated under control conditions (Figure 4). The NECA-related enhancement of the dye-uptake rate was shown to be linked to an increased expression of Cx26 [11]. We therefore assume that CVB2-61 and CVB4-57 did not negatively interfere with gene expression e.g., the expression of Cx26.

The action mechanisms of the agents CVB2-61 and CVB4-57 affecting the Cx hemichannels are not yet elucidated. For small molecules, such as long-chain alcohols and anaesthetics that inhibit Cx channels, an indirect effect due to induced changes in membrane fluidity has been proposed [56,57,58]. This may explain why these small molecules and anaesthetics do not distinguish between the gap junction channels and the Cx hemichannels. Likewise, these small molecules and anaesthetics are not expected to achieve any isoform specificity. Since the gap junction channels were not affected by neither CVB2-61 nor CVB4-57 (Figure 5b), a similar model of action to that proposed for alcohols or anaesthetics seems unlikely. Finally, some small molecules may have detrimental effects on the physiology of tissue, as revealed by the observation that GA [42] and CBX affected the barrier function (Figure 5a). The mimetic peptides, such as Gap26 and Gap27 [59], and many anti-Cx antibodies bind the extracellular domains of the Cx, and thus probably interfere with Cx hemichannel opening. Many small molecules, as well as mimetic peptides and antibodies, affect more than just the Cx hemichannels. Depending on the duration of the application, they have negative impacts on the gap junction channels. These effects on gap junction channels are spontaneous and reversible for the small molecules [45]. For mimetic peptides as well as antibodies, the effects are slow and may be related to the competition of mimetic peptides or antibodies with the docking mechanism of Cx hemichannels. With regard to the Cx isoform specificity, many anti-Cx antibodies used to close the hemichannels are mainly directed against the extracellular domains of the Cxs [45]. Similarly, many mimetic peptides were designed according to the extracellular domains of the Cxs [59]. The extracellular loops of Cxs are short and largely conserved between the Cx isoforms. In this context, the development of Cx isoform specific antibodies as well as mimetic peptides may be challenging. However, as shown by Xu et al. 2017, it is possible to develop an anti-Cx26 antibody capable of inhibiting only Cx26 [51]. Furthermore, as shown for Gap19 [48], other more specific regions of the Cx protein distinct from the extracellular domains could be considered for the design of mimetic peptides, thus allowing a better Cx-isoform specificity. The agents CVB2-61 and CVB4-57 reversibly inhibited the Cx hemichannel opening and did not affect the gap junction channels when applied for a duration of 24 h (Figure 5b). These results suggest that both CVB2-61 and CVB4-7 blocked the Cx hemichannels without competing with the docking of Cx hemichannels between the adjacent cells. Current understanding of the mode of interaction between both CVB2-61 or CVB4-57 and the Cx hemichannels still needs clarification, involving experiments far above the scope of the present report, such as crystallisation of Cx hemichannels in the presence of either agent. Finally, neither CVB2-61 nor CVB4-57 were observed to affect the barrier function of Calu-3 cells (Figure 5a), suggesting that the two compounds are non-cytotoxic. Moreover, these two small molecules may be less recognised by the immunological system. These last two aspects offer an obvious advantage in the use of these agents as tools to study the role of Cx hemichannels in the function of the epithelial barrier with the aim of developing drugs to treat diseases involving Cx hemichannels.

## 4. Materials and Methods

### 4.1. Chemical Synthesis of the Bioactive Agent CVB4-57 

#### 4.1.1. General

If not otherwise stated, the different chemicals for the synthesis of CBV4-57 were obtained from Sigma Aldrich (Sigma Aldrich S.a.r.l; Saint-Quentin-Fallavier, France). Furthermore, unless otherwise noted, all experiments were carried out under a nitrogen atmosphere. Solvents (CH_2_Cl_2_ and THF) were dried via a purification solvent system MB-SP- 800 (MBRAUN, Garching, Germany). Melting points (mp) were obtained on a on a Mettler-Toledo MP50 apparatus (Mettler-Toledo, Columbus, OH, USA)and are uncorrected. IR spectra were recorded on a Thermo Nicolet Nexus spectrometer (Thermofisher Scientific, Waltham, MA, USA). NMR spectra were recorded on Bruker Avance 300 MHz spectrometers (Bruker, Wissembourg, France). The NMR spectra were acquired in CDCl_3_, and the chemical shifts were reported in parts per million referring to CHCl_3_ (δ_H_ 7.26 for proton and δ_C_ 77.16 for carbon). Signals are described as follows: s, singlet; brs, broad signal; d, doublet; t, triplet; m, multiplet. HRMS data were recorded on a Xevo G2 QTOF instrument (Waters, Milford, MA, USA). Reactions were monitored by TLC on silica gel Alugram® Xtra SIL G/UV_254_ (Macherey-Nagel, Duren, Germany). Column chromatography was performed on Machery-Nagel silica gel 0.063-0.2 mm.

#### 4.1.2. 1-(4-((Tert-butyldimethylsilyl)oxy)phenethyl)piperazine 

To a mixture of piperazine (1.36 g, 15.86 mmol) in EtOH (5mL) at 70 °C was added dropwise a solution of (4-(2-bromoethyl)phenoxy)(tert-butyl)dimethylsilane (500 mg, 1.58 mmol) in EtOH (5 mL). The reaction mixture was stirred at 70 °C for 4h. After cooling to room temperature, saturated aqueous NaCl solution was added. The aqueous layer was extracted with CHCl_3_ and the combined organic layers were washed with brine (4x), dried over MgSO_4_, and concentrated under reduced pressure, yielding to the expected compound (469 mg, 92.7%) as a light yellow oil, which was pure enough to be used in the next step.^1^H NMR (300 MHz, CDCl_3_) δ 7.07–7.00 (m, 2H), 6.74 (d, *J* = 8.5 Hz, 2H), 3.04–2.86 (m, 4H), 2.78–2.70 (m, 2H), 2.60–2.45 (m, 6H), 0.97 (s, 9H), 0.17 (s, 6H). ^13^C NMR (75 MHz, CDCl_3_) δ 153.7, 132.8, 129.4, 119.8, 61.2, 54.2, 45.8, 32.4, 25.6, 18.0, −4.5.

#### 4.1.3. 1-(3-Bromo-4-methoxybenzyl)-4-(4-((tert-butyldimethylsilyl)oxy)phenethyl)piperazine 

To a solution of piperazine derivative 1 (469 mg, 1.46 mmol) in CH_2_Cl_2_ (15 mL) were added NEt_3_ (0.3 mL, 2.2 mmol) and 2-bromo-4-(bromomethyl)-1-methoxybenzene (430 mg, 3.54 mmol). The reaction mixture was stirred at room temperature overnight. Then, water was added, and the aqueous layer was extracted with CH_2_Cl_2_. The combined organic layers were washed with brine, dried over MgSO_4_, and concentrated under reduced pressure. The crude product was purified by silica gel column chromatography (CH_2_Cl_2_ to CH_2_Cl_2_/MeOH = 98:2) to provide compound 2 as a colourless oil (545 mg, 71.7%). ^1^H NMR (300 MHz, CDCl_3_) δ 7.52 (d, *J* = 2.1 Hz, 1H), 7.21 (dd, *J* = 8.4, 2.1 Hz, 1H), 7.08–7.00 (m, 2H), 6.84 (d, *J* = 8.4 Hz, 1H), 6.78–6.71 (m, 2H), 3.88 (s, 3H), 3.43 (s, 2H), 2.80–2.37 (m, 12H), 0.97 (s, 9H), 0.17 (s, 6H). ^13^C NMR (75 MHz, CDCl_3_) δ 154.8, 153.7, 133.8, 129.5, 129.1, 119.8, 111.5, 111.4, 77.2, 61.7, 60.6, 56.2, 53.1, 52.9, 32.8, 25.7, 18.1, −4.5. HRMS (DCI-CH_4_) *m/z* for C_26_H_40_N_2_O_2Si_Br [M+H]^+^ Calcd: 519.2042, Found: 519.2016.

#### 4.1.4. 4-(2-(4-(3-Bromo-4-methoxybenzyl)piperazin-1-yl)ethyl)phenol: CVB4-57 

To an ice-cooled solution of compound 2 (545 mg, 1.05 mmol) in anhydrous THF (15 mL) was added dropwise *n*-Bu_4_NF solution (1M in THF, 2.1 mL, 2.1 mmol). The reaction mixture was stirred at 0 °C for 4 h. Saturated aqueous NaHCO_3_ solution was added, and the aqueous layer was extracted with EtOAc. The combined organic layers were washed with saturated aqueous NaHCO_3_, water, and brine, dried over MgSO_4_, and concentrated under reduced pressure. The crude product was purified by silica gel column chromatography (CH_2_Cl_2_ to CH_2_Cl_2_/MeOH = 95:5) to provide the expected compound as a white solid (418 mg, 98.3%). mp = 151-152 °C.^1^H NMR (300 MHz, CDCl_3_) δ 7.51 (d, *J* = 2.0 Hz, 1H), 7.20 (dd, *J* = 8.4, 2.1 Hz, 1H), 7.02 (d, *J* = 8.4 Hz, 2H), 6.83 (d, *J* = 8.4 Hz, 1H), 6.70 (d, *J* = 8.4 Hz, 2H), 3.88 (s, 3H), 3.44 (s, 2H), 2.87–2.31 (m, 12H). ^13^C NMR (75 MHz, CDCl_3_) δ 155.1, 154.8, 134.2, 131.3, 131.2, 129.8, 129.5, 115.8, 111.7, 111.5, 61.8, 60.7, 56.4, 52.9, 52.6, 32.3. HRMS (DCI-CH_4_) *m/z* for C_20_H_26_N_2_O_2_Br [M]^+^ Calcd: 405.1178, Found: 405.1162. IR (KBr): ν = 3522 cm^−1^.

### 4.2. Materials

Both CVB2-61 and CVB4-57 were stored at −20 °C and were solved in DMSO and stored at −20 °C used for max. 6 months. Lucifer yellow and carbenoxolone (CBX) were purchased from Sigma-Aldrich (Sigma-Aldrich, Taufkirchen, Germany) and dissolved in sodium NaCl-bath solution without Ca^2+^, composed as follows (in mM) 145 NaCl, 5.4 KCl, 1 MgCl_2_, 5 glucose, 10 Hepes (295 mOsmol; pH: 7.4). The vehicle (for the compound CBV2-61 and CBV4-57) DMSO was added to control cells in all experiments at maximal concentrations of 0.1%. Transfection reagent FuGENE® HD was purchased from Promega (Promega, Walldorf, Germany) and used according to the transfection protocol.

### 4.3. Molecular Cloning and Transfection

For this work, the expression vectors pEF1a B1-hCx26-B2-IRES-GFP and pEF1a B1-hCx46-B2-IRES-GFP [34,60] were used and constructed by gateway cloning. The genes for human connexin (hCx) isoforms hCx26 or hCx46 were cloned in the vector pEF1a, which has an IRES element between the gateway cassette and the reporter GFP. pEF-I-GFP GX [61] was a gift from John Brigande (Addgene plasmid # 45443). The Cx gene sequence insert was placed with attB sides, shown in Table 1, into a pDONR^TM^ 221 by Gateway BP cloning. By Gateway LR cloning, the Cx gene sequence was integrated into the pEF-I-GFP. The constructs (pEF1a B1-hCx26-B2-IRES-GFP and pEF1a B1-hCx46-B2-IRES-GFP) were transformed in competent *Escherichia coli* Mach 1 cells and selected on ampicillin-containing LB, and the plasmids were extracted with the QIAprep Spin Miniprep Kit (QUIAGEN, Hilden, Germany) after protocol and controlled by sequencing. As control, the peGFP-N1 vector was used with a CMV promoter and a gfp gene sequence.

For transfection of Neuro-2A (N2A) cells (DSMZ, Braunschweig, Germany; DSMZ no.: ACC 148) and HeLa cells (DSMZ; DSMZ no.: ACC 57) with the hCx26-IRES, hCx46-IRES, or the GFP plasmids, the cells were seeded on 5 mm diameter coverslips coated with rat collagen I and cultivated until 40% confluence. The cells were transfected with 0.9 µL FuGene HD (Promega) and 0.3 µg Plasmid DNA for 24 h in 300 µL cultivating media Dulbecco’s MEM/Ham’s F-12 medium (Biochrom, Berlin, Germany), and supplemented with 10% foetal calf serum (Biochrom), 1 mg/ml penicillin, and 0.1 mg/ml streptomycin (Biochrom). After 24 h, the cells were ready for the experiments.

### 4.4. Immunofluorescence Staining

The transfected N2A cells or HeLa cells were fixed with an acetone/methanol mix (1:2) for 5 min at −20 °C and blocked with 1% bovine serum albumin (BSA) in phosphate-buffered solution (PBS) for 30 min at 37 °C. The primary anti-Cx46 antibody SantaCruz Biotechnology, Inc., Heidelberg, Germany, sc-365394) and anti-Cx26 antibody; (Alomone labs, Jerusalem, Israel ACC-212) were used. The antibodies were diluted, respectively, 1:100 and 1:200 in PBS and added to the cells overnight at 4 °C. The secondary iFluor488™-conjugated anti-rabbit and anti-mouse antibodies (16,608 and 16,528) (AAT Bioquest, Sunnyvale, CA, USA) were diluted by 1:1000 in PBS with 2 μM 4’, 6-diamino-2-phenylndole (DAPI) (Sigma-Aldrich,) for 1 h at 37 °C. The cells were washed with PBS and stored at 4 °C. The cells were imaged with an Eclipse TE2000-E inverse confocal laser scanning microscope (Nikon, Düsseldorf, Germany) with a 60× water immersion objective and the software EZ-C1 (Version 3.80, Nikon, Düsseldorf, Germany).

### 4.5. Dye-Uptake Assay

The activity of Cx hemichannels was analysed by measuring the Etd uptake. The transfected N2A cells or HeLa cells as well as the Calu-3 cells (AddexBio no.: C00116001) were cultivated on coverslips coated in rat collagen I in Dulbecco’s MEM/Ham’s F-12 medium (Biochrom, Berlin, Germany) supplemented with 10% foetal calf serum (Biochrom, Berlin, Germany), 1 mg/ml penicillin, and 0.1 mg/ml streptomycin (Biochrom, Berlin, Germany) until 40% confluence. At this stage, Calu-3 cells had grown in patches composed of 500–4000 µm^2^, with a single cell covering approximatively 100 µm^2^. The coverslips were placed in a perfusion chamber containing about 400 µL NaCl bath solution (NaCl-BS) containing 121 mM NaCl, 5.4 mM KCl, 6 mM NaHCO_3_, 5.5 mM glucose, 0.8 mM MgCl_2_, 2 mM CaCl_2_, and 25 mM HEPES (pH 7.4, 295 mOsmol/l), and mounted on an Eclipse Ti microscope (Nikon GmbH) equipped with an Orca flash 4.0 CCD camera (Hamamatsu Photonics Germany). Regions of interest (ROIs), corresponding mostly to single cells for N2A or Hela cells and to cell patches for Calu-3 cells, were selected. Thereafter, the cells were perfused with Etd containing NaCl-BS, as described above, and the recording of the fluorescence images was started. Fluorescent images were taken every 15 s with an exposure time of 900 ms, 40× or 20× objective, dependent on the cell type (N2A, HeLa 40×, Calu-3 20×). For acquisition analysis and storage of the images, NIS-Elements AR 4.4 software (Nikon GmbH) was used. For the superfusion of the different media onto the cells, the ISMATEC REGIO ICC peristaltic pump (Cole-Parmer GmbH, Wertheim, Germany) was used to maintain a constant 500 µL/min medium flow rate. After a recording time of about 3 minutes, the cells were superfused with Etd containing NaCl-BS without Ca^2+^. The agents CVB2-61 and CVB4-57 were tested in Ca^2+^-free Etd containing NaCl-BS. The rate of dye uptake was estimated by considering the dye uptake within the first minute of the respective perfusion steps.

### 4.6. Transepithelial Electrical Restistance (TEER) Measurement

For the TEER measurements, 10^5^ Calu-3 cells were seeded in transwell inserts (0.3 cm^2^) with a transparent porous PET membrane with pores of: 0.4 μm in diameter (BD Falcon, Corning, Kaiserslautern, Germany) and cultivated in the same cell culture medium as described above for 3 days. Thereafter, they were transferred into the cellZscope+ (nanoAnalytics, Muenster, Germany) and placed in the cell culture incubator for a continuous monitoring of the TEER by impedance spectroscopy. After formation of a stable barrier of more than 1000 Ωcm^2^ (approximately 5 days), the compounds CBX (100 µM), CVB2-61 (20 µM) and CVB4-57 (20 µM) were applied, and the resistance was monitored for further 24 h.

### 4.7. Gold-Nanoparticle-Mediated Laser Perforation/Dye Transfer (GNOME-LP/ DT)

GNOME-LP/ DT experiments were performed as previously described [11,42]. The cells were seeded at a density of 10^5^ cells/well in 96-multiwell plates and cultivated for 72 h until confluence. Gold nanoparticles (AuNPs, diameter 200 nm, 0.5 μg/cm^2^) were added 3 h before an experiment started. Cells were washed with a Ca^2+^-free NaCl-BS, as described above. The laser permeabilisation was performed in the presence of 0.25% Lucifer yellow (LY) dissolved in Ca^2+^-free NaCl-BS, using the laser set-up and parameters previously published [62]. In each well of a 96-multiwell plate, a line of cells was optoperforated by a 20 mW laser beam with a diameter of 60 µm and a scanning velocity of 50 mm/s. After 10 min dye diffusion time, the cells were washed with Ca^2+^-containing bath solution and then fixed with 4% formaldehyde. In some experiments, CBX (100 µM), CVB2-61 (20 µM) or CVB4-57 (20 µM) was present during optoperforation in the dye. To automatically document the GNOME LP/DT experiments, images were obtained with an Orca Flash 4.0 camera mounted onto an Eclipse Ti microscope (Nikon, Düsseldorf, Germany) with a 10× objective using the NIS elements AR 4.21 software (Nikon, Düsseldorf, Germany). The tool Multipoint ND acquisition was used to generate three images with 2044 × 2048 pixels (1348.4 × 1351.04 μm) per well along the perforated fluorescent cell band. To analyse the diffusion distance in fluorescence plot profiles, a lab-based ImageJ plugin was used, which generated a one pic of the three images, including a background subtraction and Gaussian fitting; results with a R^2^ value < 0.96 were not used.

### 4.8. Statistical Analyses

Experiments were repeated 3-9 times independently. All statistical analysis was performed using Microsoft Excel software (Microsoft Office 2016) and Origin software (OriginLab 7.0). One-way ANOVA were performed for all studies, with *p* < 0.05 as the minimal cutoff for statistical significance. The data are given as mean with the error bars to indicate the standard deviation (SD) or standard error of the mean (SEM).

## 5. Conclusions

The present paper identifies Cx and specifically Cx26 hemichannels as targets for the bioactive phenolic agents CVB2-61 and CVB4-57. The blocking of Cx hemichannels in epithelial cells could be a way to reduce the release of DAMPs, such as nucleotides, by the epithelial cells during inflammatory events, and may represent, at least partly, the mode of the anti-inflammatory effect of phenolic agents. The results show that CVB2-61 and CVB4-57 reversibly blocked the Cx26 hemichannels without affecting the gap junction communication or the barrier function of the epithelial cells. The results suggest a role of epithelial cells and Cx hemichannels in the discussed positive effects of phenolic agents in the treatment of inflammation. In this context, CVB2-61 and CVB4-57 could be a starting point for further investigation of Cx hemichannel-based therapies for different pathologies that involve Cx hemichannels.

## Figures and Tables

**Figure 1 pharmaceuticals-15-01173-f001:**
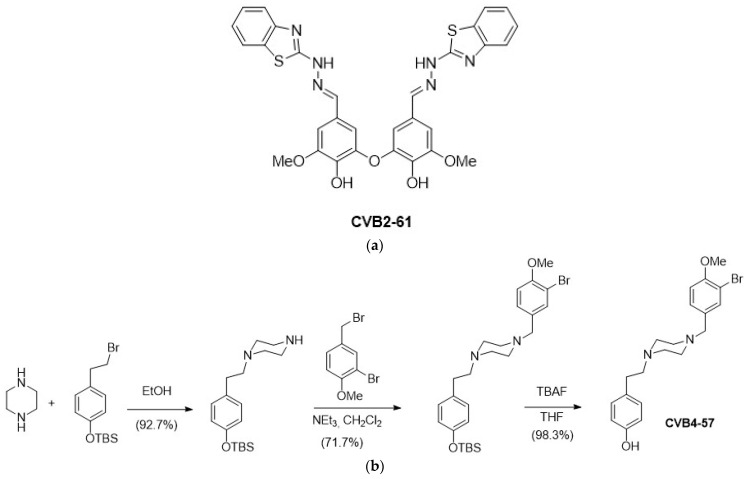
Chemical structure of the studied bioactive phenolic agents. (**a**) CVB2-61 was synthesised as previously described [29], and (**b**) CVB4-57 was synthesised in three steps, as depicted.

**Figure 2 pharmaceuticals-15-01173-f002:**
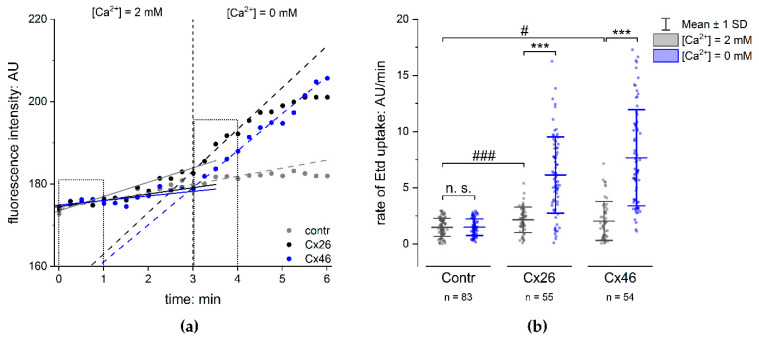
Dye uptake in N2A cells expressing connexins. (**a**) The time course of the fluorescence intensity of Etd in N2A cells transfected with GFP vector (Contr), hCx26IRES-GFP (Cx26) vector, or Cx46IRES-GFP vector (Cx46). The dots show the fluorescence intensity measured in single cells expressing only GFP (grey), Cx26 (black), and Cx46 (blue) in the presence and absence of external Ca^2+^. Removal of external Ca^2+^ led to a rapid increase in the measured fluorescence intensity in the cells expressing Cxs, but not in the control cells, indicating an opening of the Cx hemichannels. The lines show a linear regression along the fluorescence intensity in the presence and absence of external Ca^2+^; the slope was used to estimate the rate of Etd uptake. (**b**) Rate of Etd uptake in N2A cells transfected with the Cx isoforms compared to cells expressing GFP alone (Contr). The results are given as mean ± SD of at least three respective transfections. The number of analysed cells is given as n. One-way ANOVA was applied to estimate the statistical significance (*p*: # < 0.05; ***, ### < 0.001).

**Figure 3 pharmaceuticals-15-01173-f003:**
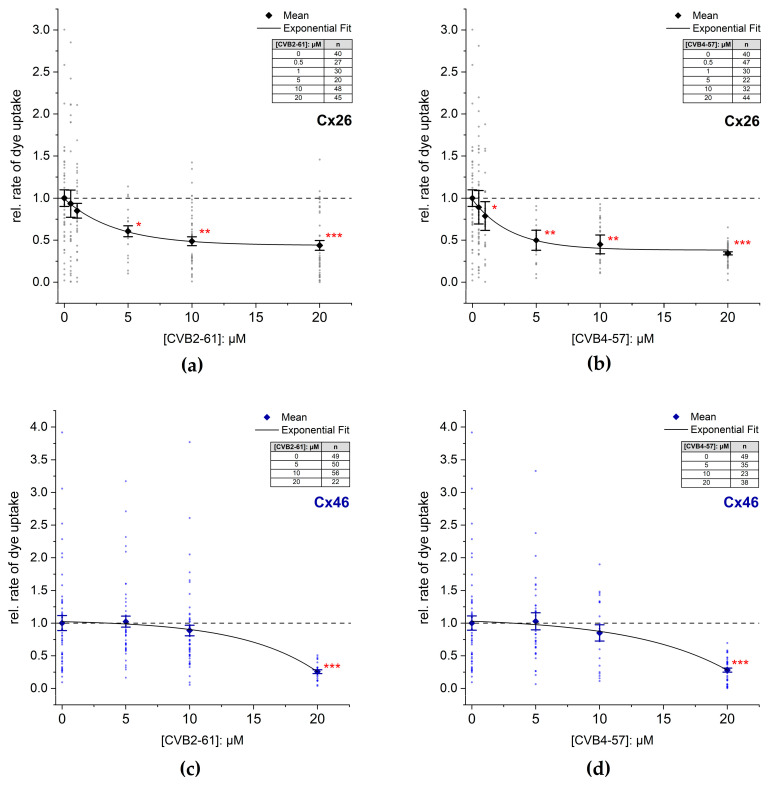
Inhibition of dye uptake by the agents CVB2-61 or CVB4-57. The rate of dye uptake in Cx26 (**a**,**b**) or Cx46 (**c**,**d**) expressing N2A cells was measured in Ca^2+^ free external solution in the presence or absence of the indicated concentrations of CVB2-61 (**a**,**c**) and CVB4-57 (**b**,**d**). Representative experiments indicating the time course of the dye uptake in the presence of the compounds are given in Appendix A. The data were normalised to the mean rate of dye uptake found in N2A cells expressing Cxs after removal of external Ca^2+^ and in the absence of the agents. The data are given as mean ± SEM for at least three respective transfections. The number of cells used for each experiment is given as n. The lines were obtained by fitting the means to a single exponential function. One-way ANOVA was applied to estimate the statistical significance to the 0 µM treated cells (*p*: * < 0.05; ** < 0.01; *** < 0.001).

**Figure 4 pharmaceuticals-15-01173-f004:**
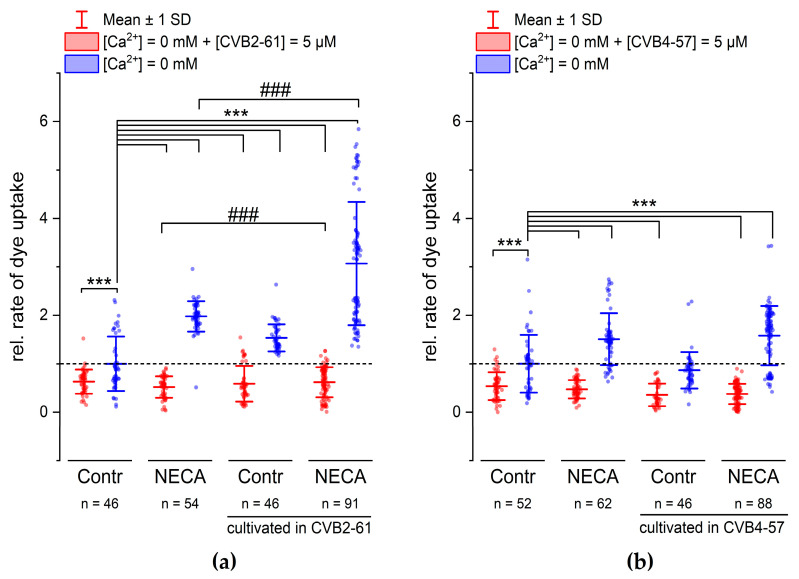
CVB2-61 and CVB4-57 inhibited the dye uptake in Calu-3 cells. The effect of the compounds (5 µM) (**a**) CVB2-61 or (**b**) CVB4-57 on the rate of dye uptake after removing of external Ca^2+^. When applied for the short period of the dye-uptake assay, both CVB2-61 and CVB4-57 maintained the reduced dye-uptake rate in cells cultivated under control conditions and those cultivated in the presence of NECA (24 h, 10 µM). The removing of CVB2-61 or CVB4-57 was followed by an increased dye-uptake rate. Note the enhanced dye-uptake rate observed in cells cultivated in the presence of NECA was similar to recently published results [11,40]. For cells cultivated with the compounds for 24 h, modifications of the dye-uptake rate were only observed for CVB2-61. Cells cultivated with CVB2-61 showed a significantly enhanced dye-uptake rate compared to cells cultivated under control conditions (Contr). Correspondingly, cells cultivated with CVB2-61 and NECA showed a more enhanced dye-uptake rate compared to cells cultivated with NECA alone. The data were normalised to the mean of the rate of dye uptake observed in cells cultivated under control conditions (Contr, blue box plot). The data are given as mean ± SD of at least three biological replicates; n indicates the number of analysed cell patches. A cell patch contained 5–40 cells. One-way ANOVA was applied to estimate the statistical significance (*p*: ***, ### < 0.001).

**Figure 5 pharmaceuticals-15-01173-f005:**
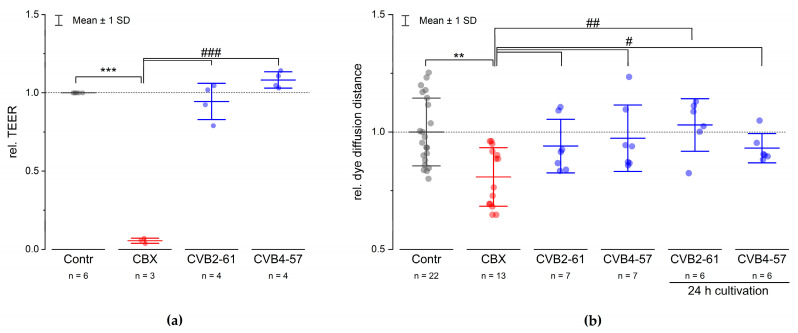
CVB2-61 and CVB4-57 did not affect the barrier function or gap junction communication. (**a**) The effect of CVB2-61 (20 µM) and CVB4-57 (20 µM) on the TEER of Calu-3 cells cultivated in transwell inserts with a TEER value of at least 1000 Ωcm^2^ is shown. (**b**) The dye diffusion distance in the Calu-3 cells monolayer as estimated by the gold-nanoparticle-mediated laser perforation/dye transfer method (GNOMELP/ DT; [42]) in presence or absence of CVB2-61 (20 µM) and CVB4-57 (20 µM). Compared to the control cells, the compounds CVB2-61 and CVB4-57 did not change either the TEER value or the dye diffusion distance. Notably, even if the compounds were already present in the cells over a cultivation time of 24 h, the dye diffusion distance was not affected. In contrast, CBX (100 µM), a commonly used inhibitor of Cx channels, strongly reduced the TEER (**a**), and as expected, the dye diffusion distance (**b**). The data were normalised to the mean of the results found in the corresponding control cells and are given as mean ± SD of at least three respective experiments; n indicates the number technical replicates. One-way ANOVA was applied to estimate the statistical significance (*p*: ***, ### < 0.001; **, ## < 0.01; # < 0.05).

**Table 1 pharmaceuticals-15-01173-t001:** Primers used for the BP cloning to generate the various entry clones [34].

Primer	5’-3’ Sequence
GW_BP-cloning hCx26 attB1 F	GGGGACAAGTTTGTACAAAAAAGCAGGCTTAATGGATTGGGGCACGCT
GW_BP-cl. hCx26 stop attB2 R	GGGACCACTTTGTACAAGAAAGCTGGGTTCTAAACTGGCTTTTTTGACTTCCCAGAAC
GW_BP-cloning hCx46 attB1 F	GGGGACAAGTTTGTACAAAAAAGCAGGCTCCATGGGCGACTGGAGCTTTCTGG
GW_BP-cl. hCx46 stop attB2 R	GGGGACCACTTTGTACAAGAAAGCTGGGTTCTAGATGGCCAAGTCCTCCGGT

## Data Availability

Data are contained within the article and Appendix A.

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
