# Peer review of "The Bioactive Phenolic Agents Diaryl Ether CVB2-61 and Diarylheptanoid CVB4-57 as Connexin Hemichannel Blockers"

_pharmaceuticals, 2022, doi:10.3390/ph15101173_

Round 1

Reviewer 1 Report

Dierks et al show a convincing experimental design to demonstrate the eficiency of two new phenolic compounds as inhibitors of Cx-26 and Cx-46 hemichannels. The novelty of this work is based on the fact that these compounds seem to not affect gap-junction function. However, some issues need to be addressed before the acceptance of this manuscript for publication.

1. The english need to be revised

2. The quality of images provided Is poor. Better images showing the GFP-Cx pattern of expression are required. 

3. Please provide representative images of the dye uptake in all the experiments.

4. Please provide the number of cells used per experimental condition in all figures.

Also provide the dot-cloud of the experiments to expose the variability of the results.

5. Figure 5 exhibits important variability in the results, please confirm that the error bars correspond to SEM and not SD. If it is the first, please increase the number of experiments to reduce SEM

6. In figure 7b, please confirm that the effect of the treatment with CBX in dye-diffusion is different to effect of the compounds studied here.

7. Please provide Statistical analyses description in the methodology

Reviewer 2 Report

This manuscript has some valuable points and several shortcomings, beginning with title which boosts CVB2-61 and CVB4-57 as "potent connexin hemichannel blocker(s)". The presented data show instead EC50 values comprised between 5 and 10 uM for the effect on Cx26, and even higher for Cx46, which is not particularly exciting. I recommend removing "potent" from the title.

ABSTRACT:

line 22, "essey", you mean "assay".

line 22, "the impact of the agents", which agents? one presumes the two compounds mentioned in the title, however these should be adequately introduced in the abstract too.

 line 24, "revesible", here and elsewhere (e.g. line 29, 272, etc.), you mean "reversibly".

 Keywords: line 32, "TEER", spell out.

 INTRODUCTION:

line 38, although valuable, Ref. [1] is obsolete and can be replaced by a more recent one, e.g. doi: 10.1016/j.tcb.2016.06.003.

 line 39, "peptide", you mean "peptides".

 line 40-42, why are you focussing on this particular aspect of gap junction-mediated communication? Your manuscript is not about the heart. You need to broaden the perspective and mention other aspects of communication between adjacent cells (of which the mentioned  doi: 10.1016/j.tcb.2016.06.003 is repleted).

 line 42-43: "Cxs form also unopposed hemichannels 42 in the membrane of individual cells", you need a Ref. here. The same one mentioned above (doi: 10.1016/j.tcb.2016.06.003) will do, otherwise there are numerous other possibilities.

 line 47: "hemichannels allow a release", you mean "the release". You need to add that hemichannels also allow influx of solutes from the extracellular ("interstitial") milieu, on which the dye uptake assay used by you is based.

 line 96, insert comma after CVB2-61.

 line 98, replace dot with comma after [28], insert comma after CVB4-57, insert comma after "analogue".

RESULTS:

The order in which the results are presented can be improved. For example, you could start with the "Synthesis of the bioactive phenolic agents CVB2-61 and CVB4-57", currently section 2.3, which could become 2.1. You could then merge Figure 1 and Figure 4, reducing the redundancy.

 There is nothing new in Fig.2, which could be removed from the main text and merged with Figure S1. In both mentioned figures, the supposedly punctuated staining due to the anti-connexin antibodies is washed out by the prevalent green fluorescence of the GFP signal. I suggest showing the 3 colour channels (blue, green and red) of each panel as separate images, in addition to the currently shown merge image.

 Line 128-129, legend of Figure 2: "Note: some cells seem to express Cxs and no GFP. This is due to the fact that the micrographs were mainly focused on Cxs". Since GFP is cytosolic, its signal should be present in any focal plane, except, perhaps, where the cells adhere to the coverslip. Please clarify.

 Line 133, "we performed dye uptake experiments [32], [33], [34]." There are hundreds of studies where dye uptake has been used to probe hemichannel functionality. The selection of these 3 Refs, therefore, appears arbitrary and biased towards self-citation (2 out of 3). The unaware reader would be better informed by reading, instead, doi: 10.1007/s00232-016-9925-y.

 line 136: state the amount of Ca2+ (2 mM), when present.

 line 141: after "experiments", add "(Figure 3a)".

 line 144: "Figure 2b", you mean Figure 3b.

 In Fig.3a, the change of slope upon removing Ca2+ appears instantaneous (on the time scale of minutes). Since you state that "The rate of dye uptake was estimated by considering the dye uptake from minute 0 to 1 and minute 3 to 4 of an experiments run", the bath exchange time interval should be considerably shorter.Did you estimate this time interval(by, e.g. dissolving fluorescein in the vehicle and monitoring is emission over time)? Please add this quantity to the text.

 Line 165: "N2A cells transfected with variants as compared", which variants?

 Line 201: You need to add a sentence to explain what Calu-3 cells are.

 Line 245-246: you should add that CBX and GA inhibit IP3-Mediated endothelial cell calcium signalling and depolarise mitochondria; add doi: 10.1111/bph.15329 to the Ref. list.

 line 251: after "transepithelial electrical resistance", add its acronym (TEER).

 line 256: "Figure 5b", you mean "Figure 7b".

DISCUSSION:

The discussion should be rewritten, it is currently unfocussed, clearly biased and contains several repetitions and factual errors.

 Line 276: "LPS" not defined.

 Line 281: "persistent", you mean "persist".

 Line 283: "COPD", not defined.

 Line 287: "Cx hemichannels could be of interest for cancer therapy", you need a REF here.

 Lines 294-296: This statement is clearly false. I can give at least one counterexample for each class of inhibitors:

- boldine (small molecule), DOI: 10.1002/glia.23182;

- Gap19 (peptide), DOI: 10.3389/fncel.2014.00306

- abEC1.1 (antibody), DOI: 10.3389/fnmol.2017.00298

 Line 301: "suggesting that the agents specifically target the Cx hemichannels (Figure 7b)", this statement is not supported. You have not tested the two agents on any other type of membrane channels, transporters, receptors, etc.

 MATERIALS AND METHODS:

 Line 423, "month", make plural, "months".

 Line 423, "Lucifer Yellow, carbenoxolone (CBX) was purchased...", you mean "Lucifer Yellow AND carbenoxolone WERE purchased...".

 Line 436, "attB sides". I presume this refers to the sequences shown in Table 1, which is not referenced in the text.

 Line 451, you need to reference Table 1 somewhere in the text.

 465, ethidium bromide. "Edt" is redefined 3 times: in the Abstract, here and again on line 475.

 CONCLUSION

 Lines 520-522, "The blocking Cx hemichannels in epithelial cells repressing thereby the release of PAMPs by the epithelial cells could be, at least partly, the mode of the anti-inflammatory effect of phenolic agents." This conclusion is not supported by the data, as you have not performed any release assay of inflammatory substances (e.g., ATP).

 Figure S1 (legend), line 536-537, "in structures probable vesicles...", you mean "probably".

Round 2

Reviewer 1 Report

All my concerns were addressed by authors. I consider that the manuscript is suitable for publication in its  actual version.